# Artificial Intelligence as A Complementary Tool for Clincal Decision-Making in Stroke and Epilepsy

**DOI:** 10.3390/brainsci14030228

**Published:** 2024-02-28

**Authors:** Smit P. Shah, John D. Heiss

**Affiliations:** 1Resident Physician, University of South Carolina School of Medicine, PRISMA Health Richland, Columbia, SC 29203, USA; 2Senior Clinician and Neurosurgical Residency Director, Surgical Neurology Branch [SNB], Building 10, Room 3D20, 10 Center Drive, Bethesda, MD 20814, USA; john.heiss@nih.gov

**Keywords:** artificial, intelligence, neurology, stroke, epilepsy, neuroimaging

## Abstract

Neurology is a quickly evolving specialty that requires clinicians to make precise and prompt diagnoses and clinical decisions based on the latest evidence-based medicine practices. In all Neurology subspecialties—Stroke and Epilepsy in particular—clinical decisions affecting patient outcomes depend on neurologists accurately assessing patient disability. Artificial intelligence [AI] can predict the expected neurological impairment from an AIS [Acute Ischemic Stroke], the possibility of ICH [IntraCranial Hemorrhage] expansion, and the clinical outcomes of comatose patients. This review article informs readers of artificial intelligence principles and methods. The article introduces the basic terminology of artificial intelligence before reviewing current and developing AI applications in neurology practice. AI holds promise as a tool to ease a neurologist’s daily workflow and supply unique diagnostic insights by analyzing data simultaneously from several sources, including neurological history and examination, blood and CSF laboratory testing, CNS electrophysiologic evaluations, and CNS imaging studies. AI-based methods are poised to complement the other tools neurologists use to make prompt and precise decisions that lead to favorable patient outcomes.

## 1. Introduction

In the coming years, the complexity of data used in Neurology’s clinical and research aspects will proliferate. Electronic medical records hold vast amounts of information. Major health systems rely on data-heavy technology to analyze clinical and genomic information. Computer analysis of digital medical data could aid the neurologist in making diagnoses, detecting disease patterns, and detecting health vulnerabilities. With its sophisticated machine learning algorithms, AI offers efficient and practical tools to clinicians to better interpret, access, and understand clinical information and narrow differential diagnoses in simple and complex cases [1,2]. AI has demonstrated great clinical utility in the management of Migraines as demonstrated by Torrente A. et al. [3]. Due to a high incidence of Stroke and Epilepsy in United States, which have been leading causes of morbidity and mortality, we would like to focus, exhibit, and discuss potential applications of AI in these two fields specifically by presenting our literature review and innovations so far, which can serve as great clinical adjuncts for clinicians which, in turn, can help deliver excellent patient care. Artificial intelligence could aid the neurology subspecialties of stroke and epilepsy by increasing the speed and consistency of analysis of clinical imaging studies and other data and clinical decision-making. Artificial intelligence can use evidence-based medicine practices to assure that the most modern and accepted medicine is being delivered. Artificial intelligence systems draw on extensive data sets of clinical information and are less prone than humans to have recency, recall, and other biases that can lead to inaccurate conclusions or ranking of the likelihood of the various diagnoses in a differential diagnosis. AI can help usher the era of personalized medicine into routine neurology clinical practice.

## 2. Basic Terminology and Concepts of AI

Key AI terms include ‘Machine Learning’, ‘Supervised Learning’, ‘Unsupervised Learning’, ‘Model and Training’, ‘Artificial Neural Network’, ‘Deep Neural Network’, ‘Convolutional Neural Network’, ‘Black Box’ and ‘Reinforcement Learning’ [4,5]. 

*Machine Learning:* Machine Learning [ML] is a field of AI associated with developing, studying, and generalizing statistical algorithms over time to perform tasks without specific instructions. A developed algorithm encodes statistical regularities extrapolated inherently from a database of examples to assess parameters for future predictions [4,6].

*Supervised Learning: *Supervised Learning [SL] uses previously established expert-labeled training examples to create an algorithm to assess parameters for future predictions. Its paradigm is analogous to machine learning because input and output values are used to train the algorithm model and derive the function relating input to output values. The SL function analyzes new data and derives the expected output values. Because SL creates a learning algorithm from training data, it may misinterpret data related to situations or diagnoses not present in the training data. SL is susceptible to errors from incomplete training data, so-called generalization errors [4,7].

*Unsupervised Learning: *Unsupervised Learning [UL] is less constrained than Supervised Learning because algorithms are learned and developed from the patterns in unlabeled data. In UL, machine learning algorithms discover patterns or data groupings without human intervention [4,8].

* Modeling and Training: *Modeling trains a machine-learning algorithm to make predictions from unseen data. Training coincides with modeling, where machine learning algorithms are fed examples from a training data set to update and calibrate parameters for future predictions. In model training, information types and their weights and bias fit into a machine learning algorithm to improve function over the predictive range [4].

*Artificial Neural Network [ANN]:* A machine learning technique that amalgamates and processes many layers of information, each holding essential parameters extracted incrementally from training data. Brain neuron network organization inspired this concept. Signals travel from input to output after traversing all layers multiple times [4,9].

*Deep Neural Network:* A deep neural network [DNN] is an artificial neural network [ANN] with multiple layers between the input and output layers. The various types of neural networks share these components: neurons, synapses, weights, biases, and functions. These components function together like brain neural networks. A DNN can be trained like other ML algorithms [4,10].

*Convolutional Neural Network:* Like the human visual cortex, the convolutional neural network displays connectivity patterns. It is a feed-forward neural network that learns feature engineering via filter optimization [4,11].

*Black Box:* Black box AI models arrive at conclusions or decisions without explaining how they were reached. The precise steps leading to the Black Box model’s predictions cannot be explained because the predictions arise from unexplained parameters being processed by a highly complex analysis maze that is machine-derived and not a direct product of human consciousness and thought processes [4]. 

### Reinforcement Learning

Reinforcement learning [RL] is a machine learning training method that develops decision algorithms by rewarding desired behaviors and punishing undesired ones. RL depends on environmental interactions. The algorithm receives rewards or penalties according to the desirability of behaviors and learns through this editing to make better decisions over time. The RL algorithm completes tasks without earlier instructions. It can learn while failing to complete the task. It derives basic rules guiding future predictions from experience performing the task [5].

## 3. Methods

To write this review, we searched PubMed using the key words “Artificial Intelligence”, “Acute Ischemic Stroke”, “Epilepsy”, “Clinical Decision Making” and “Intracranial Hemorrhage (ICH)” for articles published on these subjects between 2000 and 2023 (Figure 1). From these articles, we decided which papers utilized AI in their decision-making. Articles describing studies that answered research questions about the clinical utility of AI methods were then selected and reported in tabular format (Table 1, Table 2 and Table 3). The Quality Improvement method of the Plan-Do-Check-Act was suggested as a way for ongoing testing and improving of AI algorithms used in clinical practice (Table 4). 

## 4. Discussion

A growing body of literature suggests that artificial intelligence is becoming an invaluable tool for stroke and epilepsy clinicians. Studies report AI applications complementing traditional neurological care and improving diagnostic accuracy and clinical outcomes. As discussed above, early AI applications in the 2000s used clustering to analyze MRI sequences for regional brain perfusion properties. AI applications are standard care tools at the major level in CSCs [Comprehensive Stroke Centers] for analyzing CT perfusion studies and detecting large vessel occlusion [LVO]. The field of Stroke Neurology has improved its care systems by perfecting diagnostics and hastening stroke care. For example, AI tools can help minimize transfer time and improve outcomes by shortening the time to treatment with thrombolytics or mechanical thrombectomy. CT perfusion studies hold data critical to evaluating the cerebral vascular physiology after a stroke. A fundamental measure is rCBF [relative Cerebral Blood Flow], the flow rate through the vasculature in the brain region of interest [ROI]. Other measures include rCBV [relative Cerebral Blood Volume], the volume of blood within the ROI vasculature; MTT [Mean Transit Time], the average time for arterial-to-venous blood transit through infarcted tissue; and TTP [Time-To-Peak] the time interval between first appearance to peak enhancement of contrast-containing blood in the arterial vessels [46]. These CT perfusion imaging factors help assess the Mismatch Ratio and the infarct Core. Clinical decisions on the likelihood of improvement with mechanical thrombectomy consider these measures and the Modified Ranking Score [mRS]. AI assures clinical decisions are evidence-based, consistent with diagnostic and treatment guidelines, and give proper weight to relevant diagnostic and prognostic factors.

Acute decision-making in AIS uses AI for rapid and reliable analysis of perfusion and vessel imaging [Table 1—via PubMed search]. AI has vessel-imaging applications beyond the AIS setting. For example, in the setting of intracranial atheromatous disease or multiple vascular risk factors, AI can help predict cognitive impairment and other patient outcomes in a patient. Physicians can explore the nonemergent role of AI in vessel imaging by using Deep Convoluted Neural Networks and Generative Adversarial Networks to generate automated perfusion maps that stratify a patient’s AIS risk.

Convoluted Deep Neural Networks have been used extensively to predict the prognosis of ICH patients [Table 2—via PubMed search]. In addition, AI software can detect ICH and chronic cerebral microbleeds, ascertain ICH volume, and predict the rate of ICH expansion. AI can aid in emergency room intake neuroimaging of patients with suspected ICH. AI methods give clinicians precise volumetric and quantitative analysis of ICH’s intraparenchymal and intraventricular components, guiding treatment that may lower the morbidity and mortality of ICH in these patients. Additionally, AI analysis of serial imaging in an ICU-level setting may guide physician prognostication of ICH expansion or stability and patient outcome. Some AI studies estimate the functional outcomes of ICH patients. A physician knowing the outcome AI predicts and the relevant prognostic clinical information not considered by the AI can give patients’ families an evidence-based view of the expected ICH outcome that aids decision-making.

In Epilepsy, AI can detect ictal and interictal patterns in routine and long-term EEGs. AI-based EEG analysis can be applied to adult and pediatric epilepsy patients [Table 3—via PubMed search]. AI programs may provide clinicians with information about which AED regimen would lead to better seizure control for patients with known epilepsy syndromes or genetic mutations predisposing patients to epilepsy. Also, using AI, the risk of epileptogenicity of focal MRI lesions can be predicted by routine or 1 h EEGs. This information can guide the decision for advanced neuroimaging for epilepsy patients who are epilepsy surgery candidates. This would be key in the current era given the significant evolution of surgical application in treatment refractory epilepsy patients and severely morbid conditions leading to epilepsy including Tuberous Sclerosis and Rasmussen’s Encephalitis.

Artificial intelligence’s continued adoption in neurology depends on clinicians and researchers continuing to test and improve AI prediction models. The quality improvement models used in industry can be used to continually improve AI by reducing diagnostic and other experience-based prediction errors. As new AI methods and protocols evolve, medical experts should iteratively compare expected and actual results to judge their validity, accuracy, and clinical value. Designing an AI algorithm is a plan, or hypothesis, that the algorithm will be of clinical value. However, testing an AI algorithm allows iterative scientific hypothesis testing and revision until the hypothesis fits the data. After the final version of the algorithm fits the practice data set, the algorithm is tested with new data to assess its accuracy and error rate. After that, the algorithm is revised as necessary using quality improvement methods. The quality improvement steps are [1] Plan, [4] Do, [6] Check, and [7] Act- PDCA cycle [Table 4] [45].

A sole human clinician can only see a tiny fraction of the patients covered by an extensive healthcare system and knows his patient outcomes, those reported by his colleagues, and those reported in the clinical literature. AI can potentially draw upon data from the entire healthcare system to derive diagnostic and prognostic information that can fill gaps in a neurologist’s experience or serve as reminders before decision-making. AI can retrospectively mine data for suspected and unsuspected factors leading to an AIS or ICH that could inform future medical treatment of at-risk individuals in a neurologist’s and primary care physician’s practice.

The PDCA quality improvement cycle rigorously reviews the predicted and actual outcomes of AI-based methods, leading to their progressive updating and improvement. The AI models from practice data sets are tested with new clinical information and revised appropriately. Testing of mature AI models with new data assesses their clinical value and error rate. AI models can be revised and re-tested iteratively until their accuracy is clinically valuable. Many organizations and companies adopted the Deming PDCA cycle to improve their systems and functional outcomes. Implementing the PDCA concept can ensure AI-based protocols have continued quality improvement, regular checks to assess their outcomes, and are developed into clinically valuable and reliable products.

## 5. Conclusions

AI is a diagnostic and prognostic tool to help neurologists assess patients more efficiently and treat them more effectively. AI can usher in a new era in clinical neurology by supplying a complementary tool in stroke and epilepsy that improves diagnostics and systemic efficiency, enabling better and more predictable functional patient outcomes. From a futuristic standpoint, as more data is collected by various systems-based practices in the field of medicine, with the implementation of PDCA and more efficient AI-based stroke and epilepsy protocols, implementation systems can be utilized as adjuncts to clinical evaluation in the field of Neurology.

## Figures and Tables

**Figure 1 brainsci-14-00228-f001:**
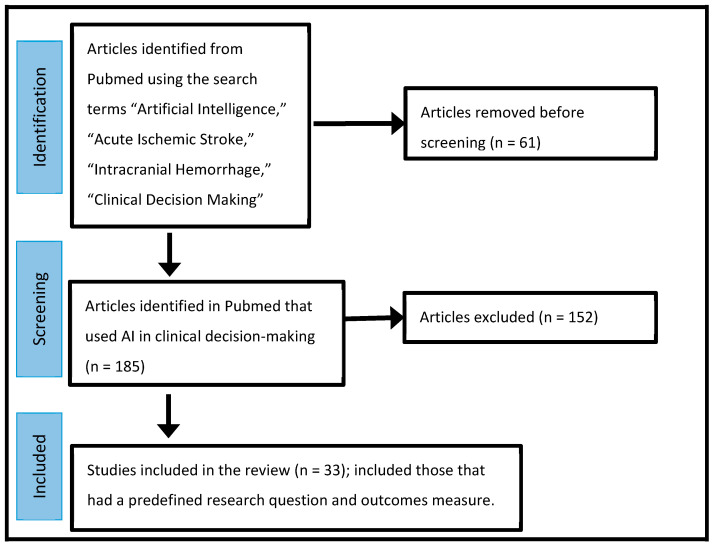
Flow diagram of the search strategy.

**Table 1 brainsci-14-00228-t001:** Summary of some studies showing the application of AI for initial neuroimaging in AIS [Acute Ischemic Stroke] between 2000 and 2023.

Year	Authors	Research Question	Outcomes Measures/Conclusions
2023 [12]	Field N. et al.	Does supplying an LVO detection algorithm notification to the thrombectomy team’s cell phone improve ischemic stroke workflow?	Transfer time and Mechanical Thrombectomy [MT] Initiation time decreased.
2023 [13]	Zhaou X. et al.	Does CTA derived from CT Perfusion [CTA-DF-CTP] give better image quality and diagnostic accuracy than traditional CTA in AIS?	CTA derived from CTA-DF-CTP had diagnostic accuracy comparable to traditional CTA and CTA-DF-CTP.
2023 [14]	Xiang et al.	Is it feasible to apply computed tomography perfusion [CTP] imaging-guided mechanical thrombectomy in acute ischemic stroke patients with LVO beyond the therapeutic time window?	NIHSS of MT group-CTP guided [at 6 h, 24 h, 7 days, and 30 days] was significantly better [*p* < 0.05]; however, infarct core volume approximation was too high or too low for this group.
2023 [15]	Du B. et al.	In patients with ICAS [Intracranial Atherosclerotic Stenosis] in the anterior circulation, is AI based on CBF [Cerebral Blood Flow] or sCoV [Spatial Coefficient of Variation] better for predicting vascular cognitive impairment?	Cognitive impairment seems better predicted by AI analysis of sCoV than CBF.
2023 [16]	Farsani S. et al.	Can AG-DCNN [Attention Gated Deep Convoluted Neural Network] predict infarct volume and size?	AG-DCNN, using only admission DWI, predicted infarct volumes at 3–7 days after stroke onset with accuracy like models using DWI and PWI.
2022 [17]	Kossen T. et al.	How can modern machine learning methods such as generative adversarial networks [GANs] automate perfusion map generation from [DSC-MR] Dynamic Susceptibility Contrasted MR in AIS on an expert level without manual validation?	DSC-MR using machine learning can speed up patient stratification by perfusion mapping in AIS.
2022 [18]	Long Le et al.	Can an advanced deep learning-based method accurately and rapidly assess collateral perfusion in AIS by automatically generating a multiphase collateral imaging map from dynamic susceptibility contrast-enhanced MR perfusion [DSC-MRP] images?	DSC-Enhanced MR Perfusion improved accuracy and sped the assessment of the collateral perfusion.
2021 [19]	Neeves G et al.	Can a machine-learning [ML] algorithm grade digital subtraction angiograms [DSA] by the mTICI scale?	ML of complete cerebral DSA predicted mTICI scores following EVT of MCA occlusions.
2020 [20]	Grosser M. et al.	In AIS patients, how do predictions of machine learning models based on local [regional] tissue susceptibility to ischemia compare with those of machine learning models based on global brain imaging?	Compared to single global machine learning models, locally trained machine learning models can lead to better prediction of lesion outcomes in AIS patients.
2019 [21]	Satish R. et al.	Can Convolutional Neural Network analysis of Multisequence MRI in AIS predict the ischemic core and penumbra?	CNN analysis experimentally confirmed local changes.
2019 [22]	Reid M. et al.	For detecting early severe ischemia, how does NCCT compare with multiphase computed tomography angiography [mCTA] regional leptomeningeal score [mCTA-rLMC] and an mCTA venous [mCTA-venous] perfusion lesion?	An assessment blinded to clinical information in patients undergoing endovascular therapy [EVT] showed that mCTA-venous more accurately detected early ischemia and predicted clinical outcomes than NCCT and the mCTA-rLMC score.
2018 [23]	Nielsen A. et al.	In AIS, can Deep Learning improve Tissue Outcome and Treatment Effect predictions?	Deep Learning improves predictions of final neurological outcome and lesion volume.
2018 [24]	Chung-Ho. et al.	Can imaging features and advanced machine learning use the TSS [Time Since Stroke] classification to characterize the Acute Ischemic Stroke Onset Time?	Demonstrates the potential benefit of using advanced machine learning methods in TSS classification.
2017 [25]	Yu. Y. et al.	Can machine learning models trained on perfusion-weighted magnetic resonance imaging [PWI] and diffusion-weighted MRI scans predict HT [hemorrhagic transformation] occurrence and location in AIS?	HT prediction was a machine-learning problem. Specifically, the model learned to extract imaging markers of HT directly from source PWI images.
2016 [26]	Tian X. et al.	Can clinically acceptable PCT [dynamic cerebral Perfusion Computed Tomography] images be created from low-dose CT images restored with a coupled dictionary learning [CDL] method in chronic and AIS patients?	CDL increased kinetic enhanced details and improved diagnostic hemodynamic parameter maps
2013 [27]	Fang R. et al.	Will the robust sparse perfusion deconvolution method [SPD] accurately estimate cerebral blood flow [CBF] in CTP performed at a low radiation dose?	SPD was superior to existing methods for CBF and helped differentiate normal and ischemic brain tissue.
2010 [28]	Mendrick A. et al.	Can the diagnostic yield of CTP in cerebrovascular diseases be expanded by combining arterial and venous segmentation and vessel-enhanced volume?	This artery and vein segmentation method was accurate for arteries and veins with normal perfusion. Combining the artery and vein segmentation with the vessel-enhanced volume produced an arteriogram and venogram, extending the diagnostic yield of CTP scans and making a CTA scan unnecessary.
2007 [29].	Meyer-Baese A. et al.	Do five unsupervised clustering techniques help analyze dynamic susceptibility contrast MRI time series?	Clustering is a valuable tool for analyzing and visualizing brain regional perfusion properties.

**Table 2 brainsci-14-00228-t002:** Studies applying AI to diagnosing and managing ICH [IntraCranial Hemorrhage] between 2000 and 2023.

Year	Authors	Research Question	Outcome Measures/Conclusions
2023 [30]	Feng H. et al.	Can AI use the GCS score, NIH stroke scale, INR, BUN, hemorrhage location, hematoma volume, modified Rankin score, and other risk factors to construct a prediction model for the prognosis of ICH at discharge, 3 months, and 12 months?	The study showed that prediction models for modified Rankin scores showed a relatively high predictive performance. Also, the study found risk factors and constructed a prediction model to predict poor modified-Rankin score outcomes and mortality at discharge, 3 months, and 12 months in ICH patients.
2023 [31]	Maghami M. et al.	Are machine learning methods for detecting ICH from non-contrast CT scans sufficiently precise to be considered acceptable diagnostic tests of accuracy [DTA]?	This meta-analysis showed that assessing noncontract CT scans using ML algorithms for detecting ICH had acceptable DTA.
2023 [32]	Vacek A. et al.	Can E-ASPECTS delineate the extent and distribution of ICH from brain CT?	AI software-Brainomix Ltd. (Oxford, UK) excellently delineated ICH extent- on stroke CTs by AI software in about 71% of cases. ICH extent was more likely to be over or underestimated when ICH was extensive, intraventricular, or extra-axial.
2023 [33]	Chen Y. et al.	Can a convolutional neural network [CNN] create a clinical imaging perfusion model predicting the short-term neurological outcomes of ICH patients?	The CNN prognostication prediction model was more effective than ICH scales in predicting neurological outcomes and ICH patients at discharge. Predictions improved slightly after including clinical data.
2023 [34]	MacIntosh B. et al.	Can Viola AI estimate the number and volume of hematoma clusters in traumatic brain injury and ICH patients?	The automated total hemorrhage volume estimate correlated with the per-participant hemorrhage cluster count. This tool may help evaluate various types of ICH in the future.
2023 [35]	Kotovich D. et al.	Did implementing a commercial artificial intelligence solution in a level 1 trauma center emergency room affect ICH’s clinical outcome?	Artificial intelligence computer-aided triage and prioritization software in the emergency room setting was associated with a significant reduction in 30 day and 120 day all-cause mortality and morbidity in ICH patients. It was also associated with a significant reduction in modified Rankin score on discharge.
2023 [36]	Li. Y. et al.	Can ML predict early perihematomal edema expansion [PHE] from non-contrast CT scan data in patients with spontaneous ICH?	This model was the best marker for predicting prior hematoma edema expansion in patients with ICH. It could predict early perihematomal edema expansion and improve the discrimination of early identification of spontaneous ICH in patients at risk of PHE expansion.
2023 [37]	Mastoukas S., et al.	What are AI methods’ reported sensitivity, specificity, and accuracy for detecting ICH and chronic cerebral microbleeds?	In 40 studies, overall sensitivity, specificity, and accuracy were more than 90% for ICH and cerebral microbleed detection. AI algorithms were developed from large data sets, volumetric analysis of imaging examinations, fine-tuning, and false-positivity reduction.
2022 [38]	Lim M. et al.	How do deep neural networks [DNN] and support vector machines [SVM] compare with clinical prognostic scores for prognosticating 30-day mortality and 90-day poor functional outcome [PFO] in spontaneous intracerebral hemorrhage [SICH]?	The SVM model performed significantly better than clinical prognostic scores in predicting 90-day PFO in SICH.
2021 [39]	Heit J. et al.	What is the accuracy of RAPID ICH, 2D/3D, a volitional neural network application designed to detect ICH, in detecting and measuring ICH volume?	Rapid ICH was highly accurate in detecting ICH and quantifying the volume of intraparenchymal and intraventricular hemorrhages.

**Table 3 brainsci-14-00228-t003:** Studies applying AI to diagnosing and managing Epilepsy between 2000 and 2023.

Year	Authors	Research Questions	Outcome Measures/Conclusions
2023 [40]	Zheng Z. et al.	Can EEG Deep Features and Machine Learning Classifiers assess and prognostically analyze KCNQ2 patients by combining the two well-trained models, RESNET-15 and RESNET-18, to extract deep features of EEG?	An outcome of 79% accuracy was reported in pediatric patients.
2023 [41]	Wang H. et al.	Can the multi-technique deep learning method WAE-Net use clinical data and multi-contrast MR imaging [T2WI and FLAIR images combined as FLAIR3 images] to forecast antiseizure medication treatment in a retrospective study involving 300 children with tuberous sclerosis complex-related epilepsy?	The hybrid technique of FLAIR3 could accurately localize tuberous sclerosis complex lesions, and the proposed method achieved the best performance [area under the curve = 0.908 and accuracy of 0.847] in the testing cohort among the compared methods.
2023 [42]	Asadi-Pooya A. et al.	Can AI machine learning methods reliably differentiate idiopathic generalized epilepsy from focal epilepsy using easily accessible and applicable clinical history and physical examination data?	This algorithm aimed at easing epilepsy classification for individuals whose epilepsy began at age 10 and older. The stacking classifier led to better results than the base classifier in general. Precision was 81%, sensitivity was 81%, and specificity was close to 77%.
2023 [43]	Tveit J. et al.	Can the artificial intelligence program SCORE-AI [Standardized Computer-based Organizing Reporting of EEG] be developed and validated to distinguish abnormal from normal EEGs, detect focal epilepsy epileptiform discharges and generalized epilepsy, and distinguish focal nonepileptiform and diffuse nonepileptiform EEGs?	SCORE-AI accuracy approached human expert-level and fully automated interpretation of routine EEGs. Accuracy was approximately 88.3%, significantly higher than the three previously published models comparing EEG interpretation to human experts.
2023 [44]	Gustavo T. et al.	In patients diagnosed with epilepsy wearing the mjn-SERAS brain activity sensor, can AI create a personalized mathematical model for the programmed recognition of oncoming seizures before they start using patient-specific EEG training data?	The AI program accurately detected pre- and interictal EEG segments in drug-resistant epilepsy patients.

**Table 4 brainsci-14-00228-t004:** PDCA [Plan-Do-Check-Act] Concept Extrapolation for AI [45].

Extrapolation of PDCA in AI
Plan Explore and discuss the question, assess the potential solution, and make use of the various machine learning models or methods as described above, set the endpoint in the objectives and goals, identify the potential metrics to use for implementation and quality measurement, prepare the action plan which includes implementation along with a potential route to reevaluate as needed.
Do Evaluate earlier models; train or retrain and test different machine learning models; assess and see if known machine learning solutions and components of the AI protocol can be improved or changed; test the overall solution to assess its integrity; review the code and filter out older ML models which did not work.
Check Monitor the model for fairness; assess for bias and variance; monitor the stability precisely to ascertain clarity and results; implement split testing of two methodologies; compare them head-to-head and assess to see which performs better.
Act The goal is standardization and continuous improvement, deploying the solution and continuing to monitor for biasing and variance, evaluating for areas of improvement in active machine learning algorithms and machine learning components, standardizing data, and features, and continuing the PDCA cycle accordingly.

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
