# Peer review of "Artificial Intelligence as A Complementary Tool for Clincal Decision-Making in Stroke and Epilepsy"

_brainsci, 2024, doi:10.3390/brainsci14030228_

Round 1

Reviewer 1 Report

Comments and Suggestions for Authors

The manuscript sets a broad context for the role of neurology in clinical decision-making and highlights the potential of AI in assisting neurologists. However, a more thorough analysis of the current landscape of AI in neurology and its implications for clinical practice is needed to enhance the depth and rigor of the paper. Here are some issues:

1.At the outset of the paper, it is imperative for the authors to contextualize the significance of clinical decision-making in disorders such as stroke and epilepsy. The author also need to discuss the potential challenges and limitations in current clinical decision-making, and briefly introduce the potential of AI as a solution.

2. In Tables 2 and 3, the authors provide a selection of relevant studies in the field from 2003 to 2023. Here, it is imperative for the authors to elucidate the literature screening process and its rationale. Subjective screening may introduce biases and inconsistencies, thereby compromising the integrity and reliability of the study findings.

3.AI encompasses a vast domain, and mere elucidation of basic concepts falls short of the requisite depth. Authors should contextualize their discussion by referencing tables within the paper, elaborating on various AI methods and algorithms employed by predecessors. It is imperative to delineate their respective strengths and limitations to provide a comprehensive understanding of the field.

4. It is recommended that the authors explore in the discussion section the integration of AI technology into clinical practice to assist healthcare professionals in making more precise diagnostic and treatment decisions.

5. The authors are encouraged to delve into the potential avenues of development for AI in the treatment of stroke and epilepsy, and to propose future research directions and potential enhancements.

6.The article lacks sufficient depth, and the specific contributions of the research remain unclear. It is advisable for the authors to summarize the main points and findings of the paper.

Author Response

1.At the outset of the paper, it is imperative for the authors to contextualize the significance of clinical decision-making in disorders such as stroke and epilepsy. The author also need to discuss the potential challenges and limitations in current clinical decision-making, and briefly introduce the potential of AI as a solution.

Response: Thank you for comments and feedback; Our goal in this paper is to explore the role of AI as adjunct in field of Neurology specifically in stroke and epilepsy rather than focusing on limitations. This has been thoroughly discussed in the discussion section in first paragraph and 4th paragraph of discussion section.

  1. In Tables 2 and 3, the authors provide a selection of relevant studies in the field from 2003 to 2023. Here, it is imperative for the authors to elucidate the literature screening process and its rationale. Subjective screening may introduce biases and inconsistencies, thereby compromising the integrity and reliability of the study findings.

Response : Thank you for the comments and feedback; Have added “Via Pubmed Search” next to each table citation in the discussion section. This was a basic literature screening process looking for relevant studies between 2003 and 2023.

3.AI encompasses a vast domain, and mere elucidation of basic concepts falls short of the requisite depth. Authors should contextualize their discussion by referencing tables within the paper, elaborating on various AI methods and algorithms employed by predecessors. It is imperative to delineate their respective strengths and limitations to provide a comprehensive understanding of the field.

Response  Thank you for the comments and feedback. This has been thoroughly addressed in tables and discussion section.

  1. It is recommended that the authors explore in the discussion section the integration of AI technology into clinical practice to assist healthcare professionals in making more precise diagnostic and treatment decisions.

Response: Thank you for the comments and feedback. This has been thoroughly addressed in discussion section.

  1. The authors are encouraged to delve into the potential avenues of development for AI in the treatment of stroke and epilepsy, and to propose future research directions and potential enhancements.

Response: Thank you for the comments and feedback. This has been thoroughly addressed in tables and discussion section. Have incorporated PDCA quality improvement in our discussion section

6.The article lacks sufficient depth, and the specific contributions of the research remain unclear. It is advisable for the authors to summarize the main points and findings of the paper.

Response: Thank you for your comments and feedback; summary of points have been discussed thoroughly in the discussion section paragraph 1,3,4.

Reviewer 2 Report

Comments and Suggestions for Authors

The manuscript attempts to classify and justify the use of artificial intelligence methods used in medicine, and, in particular, in neurology in the diagnosis of cerebrovascular disorders and epilepsy. Given that AI has been used in clinical medicine as a preliminary diagnosis since the 1990s and that quite complex robotic systems currently exist, AI methodology has found its greatest application today in brain research. In general, the manuscript can be taken as the basis for a possible publication. However, it is necessary to take into account a number of comments:

1. In the introduction, the authors attempt to present a glossary of the problem under consideration and a classification. In this case, a very limited list of cited sources is used. I see serious inaccuracies here. Especially when characterizing DNN, CNN. This section needs more attention. This also includes autoencoders, recurrent networks using LSTM cells, etc. The remaining methods listed here are general mathematical ones.

2. A large block of the manuscript is presented in the form of 4 tables. Tables can be saved. But at the same time, each table must be described with informative pictures, formulas, etc. In the same block, I recommend looking at and adding information on HCI - BCI. In this area, a lot of interesting things have been developed on the topic of AI, especially in neurology, with the use of EEG and EP.

3. The discussion overall looks good. I hope this chapter will expand after the nn 1, 2 is revised.

4. The conclusion also needs to be modified by presenting possible directions for future research.

Author Response

  1. In the introduction, the authors attempt to present a glossary of the problem under consideration and a classification. In this case, a very limited list of cited sources is used. I see serious inaccuracies here. Especially when characterizing DNN, CNN. This section needs more attention. This also includes autoencoders, recurrent networks using LSTM cells, etc. The remaining methods listed here are general mathematical ones.

Response: Thank you for comments and feedback. Goal of adding basic definitions to have a novice and an advanced reader be able to understand core concepts of AI and then delve into details about stroke and epilepsy application as mentioned in the table.

  1. A large block of the manuscript is presented in the form of 4 tables. Tables can be saved. But at the same time, each table must be described with informative pictures, formulas, etc. In the same block, I recommend looking at and adding information on HCI - BCI. In this area, a lot of interesting things have been developed on the topic of AI, especially in neurology, with the use of EEG and EP.

Response: Thank you for comments and feedback; all tables have been thoroughly discussed in the discussion section. As per visual aid goes given lot of evolving concepts it will be better if readers can go through the summary in tables and ascertain the take home points summarized in table.

  1. The discussion overall looks good. I hope this chapter will expand after the nn 1, 2 is revised.

Thank you for comments and feedback; we think that discussion is detailed enough at this point and tables have demonstrated good summary of results.

  1. The conclusion also needs to be modified by presenting possible directions for future research.

Response:  Thank you for comments and feedback, I have added the following in conclusion: “From a futuristic stand point, as more data gets collected by various systems based practices in the field of medicine, with implementation of PDCA more efficient AI based stroke and epilepsy protocols and implementation systems can be utilized as adjuncts to clinical evaluation in the field of Neurology.”

Reviewer 3 Report

Comments and Suggestions for Authors

The authors presented an interesting review about the use of AI in Stroke and Epilepsy. I believe that the manuscript is of interest for researchers and clinicians in the fields, but may need a minor revision before considering for publication. 

Here are my concerns:

INTRODUCTION

- I would suggest to widen the introduction a bit more, even talking about the unmet needs in medicine that the AI will try cover in the next future, both on the diagnostic and therapeutic side. Moreover, other examples of the use of the AI may be added. The authors could consider a mention to its use in migraine management, as one of the most prevalent disorders worldwide. Consider for that citing the following recent manuscript "Torrente A, Maccora S, Prinzi F, et al. The Clinical Relevance of Artificial Intelligence in Migraine. Brain Sci. 2024;14(1):85. doi:10.3390/brainsci14010085".

- the authors should even state why they decided to put together two different subjects (Stroke and Epilepsy) in the same review, and state the aims of the review. 

DISCUSSION

- the tables with the results should be put immediately after the subject is been reported, not before that. 

- authors should try to widen the discussion with future direction for the use of AI in the mentioned fields. 

Author Response

INTRODUCTION

- I would suggest to widen the introduction a bit more, even talking about the unmet needs in medicine that the AI will try cover in the next future, both on the diagnostic and therapeutic side. Moreover, other examples of the use of the AI may be added. The authors could consider a mention to its use in migraine management, as one of the most prevalent disorders worldwide. Consider for that citing the following recent manuscript "Torrente A, Maccora S, Prinzi F, et al. The Clinical Relevance of Artificial Intelligence in Migraine. Brain Sci. 2024;14(1):85. doi:10.3390/brainsci14010085".

Response: Thank you for the comments and feedback. I have added the following introduction “AI has demonstrated great clinical utility in management of Migraines as demonstrated by Torrente A. et. al (46). Due to high incidence and common incidence of Stroke and Epilepsy in United States, we would like to exhibit and discuss potential applications of AI in these two fields specifically by presenting our literature review and innovations so far which can serve as great clinical adjuncts.”

- the authors should even state why they decided to put together two different subjects (Stroke and Epilepsy) in the same review, and state the aims of the review. 

Response: Thank you for the comments and feedback. I have added the following introduction “AI has demonstrated great clinical utility in management of Migraines as demonstrated by Torrente A. et. al (46). Due to high incidence and common incidence of Stroke and Epilepsy in United States, we would like to exhibit and discuss potential applications of AI in these two fields specifically by presenting our literature review and innovations so far which can serve as great clinical adjuncts.”

DISCUSSION

- the tables with the results should be put immediately after the subject is been reported, not before that. 

Response: Thank you for comments and feedback, our goal is to presents tables and literature review first and discuss Stroke and Epilepsy simultaneously in the discussion section.

- authors should try to widen the discussion with future direction for the use of AI in the mentioned fields. 

Response: Thank you for comments and feedback; discussion is broad as you suggested and conclusion has been modified. PDCA application has also been discussed.

Round 2

Reviewer 1 Report

Comments and Suggestions for Authors

The issues present in the study were not adequately addressed by the authors. For instance, regarding the first issue, the expectation wasn't merely for the authors to delineate the limitations, but rather to elucidate the specific significance of AI, identify shortcomings in current clinical decision-making, and delineate how AI could ameliorate these deficiencies. For the second issue, the authors merely provided the database without furnishing a detailed protocol for paper selection, thus lacking transparency in the screening process. Other questions were also not addressed satisfactorily.

Author Response

Thank you for the comments

Attached below are edits as per your feedback:

(1) "The issues present in the study were not adequately addressed by the authors. For instance, regarding the first issue, the expectation wasn't merely for the authors to delineate the limitations, but rather to elucidate the specific significance of AI, identify shortcomings in current clinical decision-making, and delineate how AI could ameliorate these deficiencies.

Response: To address the reviewer’s concerns, in the Introduction, line 39-44, we added this information about the significance of AI in aiding shortcomings in current clinical decision-making: Artificial intelligence could aid the neurology subspecialties of stroke and epilepsy by increasing the speed and consistency of analysis of clinical imaging studies and other data and clinical decision-making. Artificial intelligence can use evidence-based medicine practices to assure that the most modern and accepted medicine is being delivered.  Artificial intelligence systems draw on extensive data sets of clinical information and are less prone than humans to have like recency, recall, and other biases that can lead to inaccurate conclusions or ranking of the likelihood of the various diagnoses in a differential diagnosis.

(2) For the second issue, the authors merely provided the database without furnishing a detailed protocol for paper selection, thus lacking transparency in the screening process. Other questions were also not addressed satisfactorily."

Response: To better explain our screening process, we added this information to the review article in lines 100 to 120:

Methods

To write this review, we searched PubMed using the key words “Artificial Intelligence,” “Acute Ischemic Stroke,” “Epilepsy,” “Clinical Decision Making” and “Intracranial Hemorrhage (ICH).” for articles published on these subjects between 2000 and 2023.  From these articles, we decided which papers utilized AI in their decision-making.  Articles describing studies that answered research questions about the clinical utility of AI methods were then selected and reported in tabular format (Tables 1-3).  The Quality Improvement method of Plan-Do-Check-Act was suggested as a way for ongoing testing and improving of AI algorithms used in clinical practice (Table 4).  

Figure 1. Flow diagram of the search strategy is also added to the paper. 

Reviewer 2 Report

Comments and Suggestions for Authors

Formally, all my remarks have been worked out.

Author Response

Thanks